# Exact solution of the Izergin-Korepin Gaudin model with the generic open boundaries

Xiaotian Xu[1,2,3], Pei Sun[2,3,4*] Xin Zhang[5†], Junpeng Cao[2,5], Tao Yang[1,2,3],

**1** Institute of Modern Physics, Northwest University, Xi'an 710127, China
**2** Peng Huanwu Center for Fundamental Theory, Xi'an 710127, China
**3** Shaanxi Key Laboratory for Theoretical Physics Frontiers, Xi'an 710127, China
**4** School of Physics, Northwest University, Xi'an 710127, China
**5** Beijing National Laboratory for Condensed Matter Physics, Institute of Physics, Chinese Academy of Sciences, Beijing 100190, China
*sunpeiphy@163.com †xinzhang@iphy.ac.cn

## 1 Abstract

**We study the Izergin-Korepin Gaudin models with both periodic and open integrable boundary conditions, which describe quantum systems exhibiting novel long-range interactions. Using the Bethe ansatz approach, we derive the eigenvalues of the Gaudin operators and the corresponding Bethe ansatz equations.**

# 1 Introduction

The Gaudin model [1] describes an important class of one-dimensional many-body systems with long-range interactions and has widespread applications in various research fields, such as condensed matter physics and high-energy physics. For example, they are relevant in the simplified BCS theory for small metallic particles [2, 3] and in the Seiberg-Witten theory of super-symmetric gauge theory [4,5]. In addition, Gaudin models provide powerful tools for constructing integral representations of solutions to the Knizhnik-Zamolodchikov equations [6–9].

The Gaudin operators with integrable boundary conditions can be constructed through a quasi-classical expansion of the inhomogeneous transfer matrix of quantum integrable models [7, 10, 11]. Within this framework, one can diagonalize the Gaudin operators once the exact solutions of the corresponding transfer matrix are derived. Following Gaudin's pioneering work, various integrable Gaudin models have been constructed and solved exactly [12–25]. Among these integrable models, the most well-studied ones are those with $U(1)$ symmetry, where the conventional Bethe ansatz approaches works well.

On the other hand, advancements in several analytical methods-such as the generalized Bethe ansatz method [26–28], the functional $T$-$Q$ relation [29–31], and the off-diagonal Bethe ansatz method [32–35]-have enabled us to solve non-trivial integrable models that lack $U(1)$ symmetry [32, 36–38]. These progress motivate us to explore novel Gaudin models and analyze their exact solutions.

In this paper, we study the Izergin-Korepin (IK) Gaudin model with periodic and open boundary conditions. The IK model plays a fundamental role in the study of non-$A$-type integrable models [39]. Exact solutions of the IK model with periodic and generic open boundaries have been constructed using the algebraic Bethe ansatz [40] and the off-diagonal Bethe ansatz [32, 38], respectively. The exactly solvable IK Gaudin model is constructed by following the standard approach, i.e., by proceeding with a quasi-classical expansion of the corresponding inhomogeneous transfer matrix [11, 21]. After some analytic calculations, we obtain the exact spectrum of the IK Gaudin model, which is parameterized by the solutions of the Bethe ansatz equations (BAEs).

The paper is organized as follows. In Section 2, we introduce the IK model with periodic boundary conditions and its exact solutions. Section 3 focuses on the construction of the IK Gaudin operators under periodic boundary conditions, and provides their solutions-including their eigenvalues and the corresponding BAEs. In Section 4, we present the IK model with open boundaries and demonstrate its exact solutions. Section 5 focuses on constructing the IK Gaudin model with open boundaries. In Section 6, we derive the eigenvalues of the open Gaudin operators and give their corresponding BAEs. The last section provides a summary of our results.

## 2 The IK model with periodic boundaries

### 2.1 Integrability of periodic IK model

The $R$-matrix of the IK model [39], associated with the simplest twisted affine algebra $A_2^{(2)}$, is given by

$$
R(u) = \left(\begin{array}{ccc|ccc|ccc}
c(u) & 0 & 0 & 0 & 0 & 0 & 0 & 0 & 0 \\
0 & b(u) & 0 & e(u) & 0 & 0 & 0 & 0 & 0 \\
0 & 0 & d(u) & 0 & g(u) & 0 & f(u) & 0 & 0 \\
\hline
0 & \bar{e}(u) & 0 & b(u) & 0 & 0 & 0 & 0 & 0 \\
0 & 0 & \bar{g}(u) & 0 & a(u) & 0 & g(u) & 0 & 0 \\
0 & 0 & 0 & 0 & 0 & b(u) & 0 & e(u) & 0 \\
\hline
0 & 0 & \bar{f}(u) & 0 & \bar{g}(u) & 0 & d(u) & 0 & 0 \\
0 & 0 & 0 & 0 & 0 & \bar{e}(u) & 0 & b(u) & 0 \\
0 & 0 & 0 & 0 & 0 & 0 & 0 & 0 & c(u)
\end{array}\right).
\tag{1}
$$

The expressions for the functions in Eq. (1) are

$$
a(u) = \sinh(u - 3\eta) - \sinh 5\eta + \sinh 3\eta + \sinh \eta, \quad b(u) = \sinh(u - 3\eta) + \sinh 3\eta,
$$

$$
c(u) = \sinh(u - 5\eta) + \sinh \eta, \quad d(u) = \sinh(u - \eta) + \sinh \eta,
$$

$$
e(u) = -2\mathrm{e}^{-\frac{u}{2}} \sinh 2\eta \cosh(\tfrac{u}{2} - 3\eta), \quad \bar{e}(u) = -2\mathrm{e}^{\frac{u}{2}} \sinh 2\eta \cosh(\tfrac{u}{2} - 3\eta),
$$

$$
f(u) = -2\mathrm{e}^{-u+2\eta} \sinh \eta \sinh 2\eta - \mathrm{e}^{-\eta} \sinh 4\eta,
\tag{2}
$$

$$
\bar{f}(u) = 2\mathrm{e}^{u-2\eta} \sinh \eta \sinh 2\eta - \mathrm{e}^{\eta} \sinh 4\eta,
$$

$$
g(u) = 2\mathrm{e}^{-\frac{u}{2}+2\eta} \sinh \tfrac{u}{2} \sinh 2\eta, \quad \bar{g}(u) = -2\mathrm{e}^{\frac{u}{2}-2\eta} \sinh \tfrac{u}{2} \sinh 2\eta.
$$

The $R$-matrix in (1) satisfies the quantum Yang-Baxter equation (QYBE) [41]

$$
R_{1,2}(u_1 - u_2)R_{1,3}(u_1 - u_3)R_{2,3}(u_2 - u_3) = R_{2,3}(u_2 - u_3)R_{1,3}(u_1 - u_3)R_{1,2}(u_1 - u_2),
\tag{3}
$$

and possesses the following properties:

$$
\text{Initial condition}: \quad R_{1,2}(0) = \kappa P_{1,2}, \quad \kappa = \sinh \eta - \sinh 5\eta,
\tag{4}
$$

$$
\text{Unitarity relation}: \quad R_{1,2}(u)R_{2,1}(-u) = c(u)c(-u) \times \mathbb{I}_{1,2},
\tag{5}
$$

$$
\text{Crossing relation}: \quad R_{1,2}(u) = V_1 R_{1,2}^{t_2}(-u + 6\eta + \mathrm{i}\pi)V_1^{-1},
$$

$$
V = \begin{pmatrix} & & -\mathrm{e}^{-\eta} \\ & 1 & \\ -\mathrm{e}^{\eta} & & \end{pmatrix},
\tag{6}
$$

$$
\text{Quasi-classical property}: \quad R(u)|_{\eta \to 0} = \sinh u \times \mathbb{I},
\tag{7}
$$

where $\mathbb{I}$ is the identity matrix, $P_{1,2}$ is the permutation operator, $R_{2,1}(u) = P_{1,2}R_{1,2}(u)P_{1,2}$, and the superscript $t_i$ indicates the transposition in the $i$-th space.

In the framework of the algebraic Bethe ansatz method [41], one can construct the transfer matrix

$$
t^{(p)}(u) = \mathrm{tr}_0\{R_{0,N}(u - \theta_N)R_{0,N-1}(u - \theta_{N-1})\cdots R_{0,1}(u - \theta_1)\},
\tag{8}
$$

where $\{\theta_1, \ldots, \theta_N\}$ is a set of inhomogeneous parameters. Here and below, the superscript $(p)$ indicates that the system is under the periodic boundary conditions.

By using the QYBE (3) repeatedly, one can demonstrate that the transfer matrices with different spectral parameters commute with each other [42] :

$$[t^{(p)}(u), \ t^{(p)}(v)] = 0. \tag{9}$$

The transfer matrix $t^{(p)}(u)$ acts as the generating functional of the conserved quantities of the system and the integrability of the system is thus proved.

## 2.2 Exact solutions of periodic IK model

Introduce some functions

$$\tilde{\mathbf{a}}(u) = \prod_{l=1}^{N} c(u - \theta_l), \tag{10}$$

$$\tilde{\mathbf{d}}(u) = \prod_{l=1}^{N} d(u - \theta_l), \tag{11}$$

$$\tilde{\mathbf{b}}(u) = \prod_{l=1}^{N} b(u - \theta_l). \tag{12}$$

With the help of the conventional Bethe ansatz method, the eigenvalues of the transfer matrix $t^{(p)}(u)$ can be parameterize by the following $T$-$Q$ relation [40,43]

$$\Lambda^{(p)}(u) = \tilde{\mathbf{a}}(u)\frac{\widetilde{Q}(u + 4\eta)}{\widetilde{Q}(u)} + \tilde{\mathbf{d}}(u)\frac{\widetilde{Q}(u - 6\eta + i\pi)}{\widetilde{Q}(u - 2\eta + i\pi)} + \tilde{\mathbf{b}}(u)\frac{\widetilde{Q}(u - 4\eta)\widetilde{Q}(u + 2\eta + i\pi)}{\widetilde{Q}(u - 2\eta + i\pi)\widetilde{Q}(u)}, \tag{13}$$

where

$$\widetilde{Q}(u) = \prod_{j=1}^{M} \sinh\left(\frac{u - \lambda_j - 2\eta}{2}\right), \tag{14}$$

and $M = 0, 1, \ldots, N$. The Bethe roots $\{\lambda_1, \ldots, \lambda_M\}$ in Eq. (14) satisfy the following Bethe ansatz equations (BAEs)

$$\prod_{l=1}^{N} \frac{\sinh\left(\frac{\lambda_j - \theta_l - 2\eta}{2}\right)}{\sinh\left(\frac{\lambda_j - \theta_l + 2\eta}{2}\right)} = -\frac{\widetilde{Q}(\lambda_j - 2\eta)\widetilde{Q}(\lambda_j + 4\eta + i\pi)}{\widetilde{Q}(\lambda_j + 6\eta)\widetilde{Q}(\lambda_j + i\pi)}, \quad j = 1, \ldots, M. \tag{15}$$

# 3 IK Gaudin model with periodic boundaries and its exact solutions

## 3.1 Construction of Gaudin operators

The IK Gaudin operators $\{H_1^{(p)}, \ldots, H_N^{(p)}\}$ with periodic boundary conditions can be constructed by expanding the transfer matrix at the point $u = \theta_j$ and around $\eta = 0$ as follows

$$t^{(p)}(\theta_j) = \kappa\, \mathsf{t}_0^{(p)}(\theta_j)\Big(\mathbb{I} + \eta H_j^{(p)} + \cdots\Big), \quad j = 1, \ldots, N, \tag{1}$$

$$H_j^{(p)} = \frac{\partial \ln(t^{(p)}(\theta_j)/\kappa)}{\partial \eta}\bigg|_{\eta=0}. \tag{2}$$

95 From Refs. [32, 35], we know that

$$t^{(p)}(\theta_j) = \kappa R_{j,j-1}(\theta_j - \theta_{j-1}) \cdots R_{j,1}(\theta_j - \theta_1) R_{j,N}(\theta_j - \theta_N) \cdots R_{j,j+1}(\theta_j - \theta_{j+1}). \quad (3)$$

96 The quasi-classical property of the $R$-matrix shown in Eq. (7) allows us to introduce the
97 corresponding classical $r$-matrix $r(u)$

$$R(u) = \sinh u \times \mathbb{I} + \eta\, r(u) + O(\eta^2), \qquad \text{when} \quad \eta \to 0,$$

$$r(u) = \left. \frac{\partial R(u)}{\partial \eta} \right|_{\eta=0}. \quad (4)$$

98 The matrix representation of $r(u)$ is

$$r(u) = \begin{pmatrix} c'(u) & 0 & 0 & 0 & 0 & 0 & 0 & 0 & 0 \\ 0 & b'(u) & 0 & e'(u) & 0 & 0 & 0 & 0 & 0 \\ 0 & 0 & d'(u) & 0 & g'(u) & 0 & f'(u) & 0 & 0 \\ 0 & \bar{e}'(u) & 0 & b'(u) & 0 & 0 & 0 & 0 & 0 \\ 0 & 0 & \bar{g}'(u) & 0 & a'(u) & 0 & g'(u) & 0 & 0 \\ 0 & 0 & 0 & 0 & 0 & b'(u) & 0 & e'(u) & 0 \\ 0 & 0 & \bar{f}'(u) & 0 & \bar{g}'(u) & 0 & d'(u) & 0 & 0 \\ 0 & 0 & 0 & 0 & 0 & \bar{e}'(u) & 0 & b'(u) & 0 \\ 0 & 0 & 0 & 0 & 0 & 0 & 0 & 0 & c'(u) \end{pmatrix}. \quad (5)$$

99 The non-zero entries in Eq. (5) read

$$a'(u) = -3\cosh u - 1, \quad b'(u) = 3 - 3\cosh u, \quad c'(u) = 1 - 5\cosh u,$$
$$d'(u) = 1 - \cosh u, \quad e'(u) = -4\mathrm{e}^{-\frac{u}{2}}\cosh\tfrac{u}{2}, \quad \bar{e}'(u) = -4\mathrm{e}^{\frac{u}{2}}\cosh\tfrac{u}{2},$$
$$f'(u) = -4, \quad \bar{f}'(u) = -4, \quad g'(u) = 4\mathrm{e}^{-\frac{u}{2}}\sinh\tfrac{u}{2}, \quad \bar{g}'(u) = -4\mathrm{e}^{\frac{u}{2}}\sinh\tfrac{u}{2}. \quad (6)$$

100 With the help of Eq. (7), we can obtain the expression for $\mathsf{t}_0^{(p)}(\theta_j)$ and the corresponding
101 Gaudin operators $H_j^{(p)}$

$$\mathsf{t}_0^{(p)}(\theta_j) = \prod_{l \neq j}^{N} \sinh(\theta_j - \theta_l) \times \mathbb{I}, \quad (7)$$

$$H_j^{(p)} = \sum_{l \neq j}^{N} \Gamma_{j,l}(\theta_j, \theta_l), \quad \Gamma_{j,l}(\theta_j, \theta_l) = \frac{r_{j,l}(\theta_j - \theta_l)}{\sinh(\theta_j - \theta_l)}. \quad (8)$$

102 Here $\Gamma_{j,l}(\theta_j, \theta_l)$ describes a long-range two-site interactions between sites $j$ and $l$ (with
103 $l \neq j$), which only depends on the inhomogeneous parameters $\theta_j$ and $\theta_l$. We can use the
104 spin-1 operators $S^\alpha$, $\alpha = \pm, z$ to expand the operator $r_{j,l}(u)$ as follows

$$r_{j,l}(u) = -(3\cosh u + 1)\mathbb{I}_{j,l} + 4[(S_j^z)^2 + (S_l^z)^2] - 6(S_j^z)^2(S_l^z)^2 - 2\cosh u\, S_j^z S_l^z$$
$$- (S_j^+)^2(S_l^-)^2 - (S_j^-)^2(S_l^+)^2 - 2\mathrm{e}^{-\frac{u}{2}}\cosh(\tfrac{u}{2})(S_j^z S_j^+ S_l^- S_l^z + S_j^+ S_j^z S_l^z S_l^-)$$
$$- 2\mathrm{e}^{-\frac{u}{2}}\sinh(\tfrac{u}{2})(S_j^z S_j^+ S_l^z S_l^- + S_j^+ S_j^z S_l^- S_l^z) - 2\mathrm{e}^{\frac{u}{2}}\cosh(\tfrac{u}{2})(S_j^- S_j^z S_l^z S_l^+$$
$$+ S_j^z S_j^- S_l^+ S_l^z) + 2\mathrm{e}^{\frac{u}{2}}\sinh(\tfrac{u}{2})(S_j^- S_j^z S_l^+ S_l^z + S_j^z S_j^- S_l^z S_l^+). \quad (9)$$

105 One see that the operator $r(u)$ is Hermitian when $\mathrm{i}u \in \mathbb{R}$. As a consequence, a family of
106 Hermitian operators $\{\mathrm{i}H_j^{(p)}\}$ can be obtained if the parameters $\{\theta_j\}$ all lie on the imaginary
107 axis.

108 Based on the expansion of $t^{(p)}(\theta_j)$ with respect to $\eta$ (1) and the commutation relation
109 of the transfer matrix with different spectral parameters (9), we can prove that $\{H_j^{(p)}\}$
110 mutually commute. The proof is as follows.

111 *Proof.* For convenience, we omit the symbol $(p)$ from $\mathsf{t}_0^{(p)}(\theta_j)$ and $H_j^{(p)}$ in the proof. From
112 Eq. (7), we see that $\mathsf{t}_0(\theta_j)$ is proportional to the identity matrix and commutes with any
113 operators. The commutation relation $[t(\theta_j),\, t(\theta_l)] = 0$ then leads to

$$
\begin{aligned}
&[\mathbb{I} + \eta H_j + \eta^2 H_j^{(2)} + \cdots,\, \mathbb{I} + \eta H_l + \eta^2 H_l^{(2)} + \cdots] \\
&= [\mathbb{I},\, \mathbb{I}] + \eta\Big\{[\mathbb{I},\, H_l] + [H_j,\, \mathbb{I}]\Big\} \\
&\quad + \eta^2\Big\{[H_j^{(2)},\, \mathbb{I}] + [\mathbb{I},\, H_l^{(2)}] + [H_j,\, H_l]\Big\} + \cdots \\
&= \eta^2 [H_j,\, H_l] + \eta^3(\cdots) + \cdots = 0,
\end{aligned}
\tag{10}
$$

114 Since $\eta$ is arbitrary, the coefficients of each power of $\eta$ in (10) must be zero. On examining
115 the $\eta^2$ term specifically, this yields the following equation

$$
[H_j,\, H_l] = 0.
\tag{11}
$$

116 $\qquad\square$

117 The aforementioned proof is also valid for the open system. It should be remarked
118 that we require $\mathsf{t}_0^{(p)}(\theta_j) = \lim_{\eta\to 0} t(\theta_j)/\kappa$ to be proportional to the identity operator. This
119 condition is automatically satisfied in the periodic system. However, for the open system,
120 additional constraints on the model parameters are mandated so that the condition holds
121 (see Eq. (3)).

## 3.2 Exact solutions of Gaudin operators

123 The Gaudin operator $H_j^{(p)}$ is exactly solvable. We can derive the eigenvalues of the Gaudin
124 operators directly from the $T$-$Q$ relation of the transfer matrix (13). The Bethe roots in
125 the $T$-$Q$ relation are also functions of the parameter $\eta$. The Bethe roots $\{\lambda_j|_{j=1,\dots,M}\}$ can
126 be expanded in terms of $\eta$ as follows:

$$
\lambda_j = \mu_j + \eta\nu_j + O(\eta^2).
\tag{12}
$$

127 By setting $u = \theta_j$ in the $T$-$Q$ relation (13) and taking the first derivative of $\ln\Lambda^{(p)}(\theta_j)$
128 with respect to $\eta$ at $\eta = 0$, we finally can get the eigenvalue of the periodic IK Gaudin
129 operator $E_j^{(p)}$, which read

$$
\begin{aligned}
E_j^{(p)} &= \left.\frac{\partial\ln(\Lambda^{(p)}(\theta_j)/\kappa)}{\partial\eta}\right|_{\eta=0} \\
&= \sum_{k\neq j}^{N} \frac{1 - 5\cosh(\theta_j - \theta_k)}{\sinh(\theta_j - \theta_k)} + 2\sum_{k=1}^{M} \coth\left(\frac{\theta_j - \mu_k}{2}\right).
\end{aligned}
\tag{13}
$$

130 The Bethe roots $\{\mu_j\}$ in Eq. (13) should satisfy the following BAEs

$$
\begin{aligned}
&\sum_{l\neq j}^{M} \left[-2\coth\left(\frac{\mu_j - \mu_l}{2}\right) + \tanh\left(\frac{\mu_j - \mu_l}{2}\right)\right] \\
&+ \sum_{l=1}^{N} \coth\left(\frac{\mu_j - \theta_l}{2}\right) = 0, \qquad j = 1,\dots,M.
\end{aligned}
\tag{14}
$$

131   From Eq. (9), we observe that the total $z$-component spin operator $\sum_j S_j^z$ commutes
132   with the Gaudin operator $H_j^{(p)}$ and these operators share the same eigenstates. The integer
133   $M$ now is a conserved charge representing the total number of spinons. When $M = 0$, the
134   energy is $\sum_k \frac{1-5\cosh(\theta_j-\theta_k)}{\sinh(\theta_j-\theta_k)}$, which corresponds to the vacuum state $|1\rangle^{\otimes N}$.
135   Due to the $\mathbb{Z}_2$ symmetry, all eigenstates in the $M \neq N$ sectors exhibit degeneracy.
136   Therefore, we only need to solve the BAEs (14) with $0 \leq M \leq N$. Some numerical
solutions of Eq. (14) for small-scale systems are presented in Table 1.

| $\mu_1$ | $\mu_2$ | $\mu_3$ | $iE_2^{(p)}$ | $d$ |
|---|---|---|---|---|
| 0.0000+0.4353i | 0.9196−1.5234i | −0.9196−1.5234i | −12.9430 | 1 |
| 0.0000−1.5502i | 0.0000+0.4521i | − | −12.7628 | 2 |
| −1.3935+1.5603i | 0.0000+0.5509i | 1.3935+1.5603i | −12.5338 | 1 |
| 0.0000+0.5733i | 0.0000+1.5294i | − | −12.3989 | 2 |
| 0.0000+0.5056i | − | − | −12.0349 | 2 |
| 0.0000−0.1551i | 0.0000+0.5085i | 0.0000−2.9649i | −0.1070 | 1 |
| 0.0000−1.2447i | 0.0000+3.3197i | 0.0000+1.5966i | 0.1198 | 1 |
| 0.0000+1.5035i | 0.0000−1.1513i | − | 0.1204 | 2 |
| 2.4667−2.9649i | −2.4667−2.9649i | 0.0000−2.9650i | 0.1354 | 1 |
| 1.1525−2.9649i | −1.1525−2.9649i | − | 0.1358 | 2 |
| 0.0000−2.9650i | − | − | 0.1373 | 2 |
| − | − | − | 0.1407 | 2 |
| 0.0000−0.1522i | − | − | 12.0709 | 2 |
| 0.0000−0.2218i | 0.0000−1.1745i | − | 12.4479 | 2 |
| 1.3948−1.2064i | 0.0000−0.1987i | −1.3948−1.2064i | 12.5850 | 1 |
| 0.0000+1.9031i | 0.0000−0.0970i | − | 12.7716 | 2 |
| 0.0000−0.0796i | −0.9210+1.8756i | 0.9210+1.8756i | 12.9496 | 1 |

Table 1: Numeric solutions of Eq. (14) with $\{\theta_1, \theta_2, \theta_3\} = \{-0.40i, 0.18i, 0.75i\}$. The eigenvalue of $iH_2^{(p)}$ given by Eq. (13) matches the exact diagonalization results. Here $d$ represents the degeneracy of the energy level.

137

# 4   The IK model with open boundaries

## 4.1   Integrability of open IK model

140   For an integrable system with open boundaries, in addition to the $R$-matrix, we also
141   require the boundary−related $K$-matrices [42]. In this paper, we consider the type II

142    non-diagonal $K$-matrices in Ref. [44]

$$K^-(u) = \begin{pmatrix} 1 + 2e^{-u-\epsilon}\sinh\eta & 0 & 2e^{-\epsilon+\sigma}\sinh u \\ 0 & 1 - 2e^{-\epsilon}\sinh(u-\eta) & 0 \\ 2e^{-\epsilon-\sigma}\sinh u & 0 & 1 + 2e^{u-\epsilon}\sinh\eta \end{pmatrix}, \quad (1)$$

$$K^+(u) = \mathcal{M}K^-(-u + 6\eta + i\pi)\Big|_{(\epsilon,\sigma)\to(\epsilon',\sigma')}, \quad (2)$$

$$\mathcal{M} = \text{diag}\Big\{ e^{2\eta}, \ 1, \ e^{-2\eta} \Big\}. \quad (3)$$

143    The matrices $K^-(u)$ and $K^+(u)$ satisfy the reflection equation (RE) and the dual RE
144 respectively [45, 46], specifically as follows

$$\begin{aligned} &R_{1,2}(u_1 - u_2)K_1^-(u_1)R_{2,1}(u_1 + u_2)K_2^-(u_2) \\ &= K_2^-(u_2)R_{1,2}(u_1 + u_2)K_1^-(u_1)R_{2,1}(u_1 - u_2), \quad (4) \\ &R_{1,2}(u_2 - u_1)K_1^+(u_1)\mathcal{M}_1^{-1}R_{2,1}(-u_1 - u_2 + 12\eta)\mathcal{M}_1 K_2^+(u_2) \\ &= K_2^+(u_2)\mathcal{M}_2^{-1}R_{1,2}(-u_1 - u_2 + 12\eta)\mathcal{M}_2 K_1^+(u_1)R_{2,1}(u_2 - u_1). \quad (5) \end{aligned}$$

145    Then the double-row transfer matrix of the IK model is constructed

$$\begin{aligned} t(u) = \text{tr}_0\{ &K_0^+(u)R_{0,N}(u - \theta_N)R_{0,N-1}(u - \theta_{N-1})\cdots R_{0,1}(u - \theta_1) \\ &\times K_0^-(u)R_{1,0}(u + \theta_1)R_{2,0}(u + \theta_2)\cdots R_{N,0}(u + \theta_N) \}. \quad (6) \end{aligned}$$

146    With the help of QYBE (3) and (dual) REs (4) and (5), one can prove that the transfer
147 matrices with different spectral parameters commute with each other [42] :

$$[t(u), \, t(v)] = 0. \quad (7)$$

148 This ensures the integrability of the system. The transfer matrix (6) indeed does depend
149 on the inhomogeneous parameters $\{\theta_j\}$ and four free boundary parameters $\{\epsilon, \sigma, \epsilon', \sigma'\}$.

## 4.2   Inhomogeneous $T$-$Q$ relation

151 In Refs. [32, 38], the transfer matrix $t(u)$ defined in (6) has been exactly diagonalized via
152 the off-diagonal Bethe ansatz approach. Let us recall the $T$-$Q$ relation.
153    First, introduce some functions

$$\mathbf{a}(u) = \prod_{l=1}^{N} c(u - \theta_l)c(u + \theta_l) \prod_{\alpha=\epsilon,\epsilon'} (1 - 2e^{-\alpha}\sinh(u - \eta)) \frac{\sinh(u - 6\eta)\cosh(u - \eta)}{\sinh(u - 2\eta)\cosh(u - 3\eta)}, \quad (8)$$

$$\mathbf{d}(u) = \prod_{l=1}^{N} d(u - \theta_l)d(u + \theta_l) \prod_{\alpha=\epsilon,\epsilon'} (1 - 2e^{-\alpha}\sinh(u - 5\eta)) \frac{\sinh u \cosh(u - 5\eta)}{\sinh(u - 4\eta)\cosh(u - 3\eta)}, \quad (9)$$

$$\mathbf{b}(u) = \prod_{l=1}^{N} b(u - \theta_l)b(u + \theta_l) \prod_{\alpha=\epsilon,\epsilon'} (1 + 2e^{-\alpha}\sinh(u - 3\eta)) \frac{\sinh u \sinh(u - 6\eta)}{\sinh(u - 2\eta)\sinh(u - 4\eta)}, \quad (10)$$

$$\mathbf{c}(u) = 4^{1-N}c_0 \sinh u \sinh(u - 6\eta) \prod_{l=1}^{N} c(u - \theta_l)c(u + \theta_l)d(u - \theta_l)d(u + \theta_l), \quad (11)$$

154    where $c(u)$, $b(u)$ and $d(u)$ are the non-zero elements of the $R$-matrix defined in (2).

The eigenvalue of the transfer matrix $t(u)$, denoted as $\Lambda(u)$, can be parameterized by the following $T$-$Q$ relation [32,38]

$$\Lambda(u) = \mathbf{a}(u)\frac{Q_1(u+4\eta)}{Q_2(u)} + \mathbf{d}(u)\frac{Q_2(u-6\eta+\mathrm{i}\pi)}{Q_1(u-2\eta+\mathrm{i}\pi)} + \mathbf{b}(u)\frac{Q_1(u+2\eta+\mathrm{i}\pi)Q_2(u-4\eta)}{Q_2(u-2\eta+\mathrm{i}\pi)Q_1(u)}$$

$$+ \frac{1}{\cosh(u-3\eta)}\left[\frac{\mathbf{c}(u)Q_1(u+2\eta+\mathrm{i}\pi)}{Q_1(u)Q_2(u)} - \frac{\mathbf{c}(-u+6\eta+\mathrm{i}\pi)Q_2(u-4\eta)}{Q_1(u-2\eta+\mathrm{i}\pi)Q_2(u-2\eta+\mathrm{i}\pi)}\right]. \quad (12)$$

The function $Q_i(u)$ in Eq. (12) depends on $\bar{N} = 4N-2$ parameters $\{\lambda_j | j = 1, \ldots, \bar{N}\}$

$$Q_1(u) = \prod_{k=1}^{\bar{N}} \sinh\left(\frac{u-\lambda_k-2\eta}{2}\right), \quad (13)$$

$$Q_2(u) = \prod_{k=1}^{\bar{N}} \sinh\left(\frac{u+\lambda_k-2\eta}{2}\right), \quad (14)$$

and the constant $c_0$ is specified as follows

$$c_0 = -2\mathrm{e}^{-\epsilon-\epsilon'}\left\{\frac{\cosh(\sigma'-\sigma+2\eta) - \cosh(\bar{N}\eta - \sum_{j=1}^{\bar{N}}\lambda_j)}{\cosh(\frac{\bar{N}\eta}{2} - \frac{1}{2}\sum_{j=1}^{\bar{N}}\lambda_j)}\right\}. \quad (15)$$

The analyticity of $\Lambda(u)$ requires that the residues of $\Lambda(u)$ at $u = \lambda_j + 2\eta$, $j = 1 \ldots, \bar{N}$ must vanish, which leads to the following BAEs

$$\frac{(1 + 2\mathrm{e}^{-\epsilon}\sinh(\lambda_j-\eta))(1 + 2\mathrm{e}^{-\epsilon'}\sinh(\lambda_j-\eta))\cosh(\lambda_j-\eta)}{4\sinh\lambda_j\sinh(\lambda_j-2\eta)}$$

$$= -\prod_{l=1}^{N}\sinh\left(\frac{\lambda_j-\theta_l-2\eta}{2}\right)\sinh\left(\frac{\lambda_j+\theta_l-2\eta}{2}\right)\cosh\left(\frac{\lambda_j-\theta_l}{2}\right)\cosh\left(\frac{\lambda_j+\theta_l}{2}\right)$$

$$\times \frac{c_0\, Q_2(\lambda_j+\mathrm{i}\pi)}{Q_2(\lambda_j-2\eta)Q_2(\lambda_j+2\eta)}, \qquad j = 1, \ldots, \bar{N}. \quad (16)$$

Due to the broken of $U(1)$ symmetry, the $T$-$Q$ relation in Eq. (12) includes an inhomogeneous term, which leads to significantly more complex BAEs (16) compared to those (15) in the periodic case. However, under specific conditions, the inhomogeneous term in (12) can vanish, as demonstrated in Section 4.3.

## 4.3 Homogeneous $T$-$Q$ relation

**Constrained non-diagonal boundaries** Under the following constraints

$$\mathrm{e}^{\sigma'-\sigma} = \mathrm{e}^{-4\mathbf{k}\eta}, \quad \mathbf{k} \in \mathbb{Z}, \quad (17)$$

the spectrum of the transfer matrix can be parameterized by the following homogeneous $T$-$Q$ relation [32,38]

$$\Lambda(u) = \mathbf{a}(u)\frac{Q(u+4\eta)}{Q(u)} + \mathbf{d}(u)\frac{Q(u-6\eta+\mathrm{i}\pi)}{Q(u-2\eta+\mathrm{i}\pi)} + \mathbf{b}(u)\frac{Q(u+2\eta+\mathrm{i}\pi)Q(u-4\eta)}{Q(u-2\eta+\mathrm{i}\pi)Q(u)}, \quad (18)$$

where the resulting function $Q(u)$ is

$$Q(u) = \prod_{j=1}^{M} \sinh\left(\frac{u-\lambda_j-2\eta}{2}\right)\sinh\left(\frac{u+\lambda_j-2\eta}{2}\right). \quad (19)$$

170  Here $M$ is a non-negative integer and takes the following values

$$
M = \begin{cases} N - \mathbf{k}, & \mathbf{k} \le -N, \\ N + \mathbf{k} + 1, & \mathbf{k} \ge N + 1, \\ N - \mathbf{k}, & 1 - N \le \mathbf{k} \le N, \\ N + \mathbf{k} - 1, & 1 - N \le \mathbf{k} \le N. \end{cases} \tag{20}
$$

171  The resulting BAEs now read

$$
\prod_{l=1}^{N} \frac{\sinh\left(\frac{\lambda_j - \theta_l - 2\eta}{2}\right) \sinh\left(\frac{\lambda_j + \theta_l - 2\eta}{2}\right)}{\sinh\left(\frac{\lambda_j - \theta_l + 2\eta}{2}\right) \sinh\left(\frac{\lambda_j + \theta_l + 2\eta}{2}\right)} \prod_{\alpha = \epsilon, \epsilon'} \frac{(1 - 2\mathrm{e}^{-\alpha} \sinh(\lambda_j + \eta))}{(1 + 2\mathrm{e}^{-\alpha} \sinh(\lambda_j - \eta))}
$$

$$
= -\frac{\sinh(\lambda_j + 2\eta) \cosh(\lambda_j - \eta)}{\sinh(\lambda_j - 2\eta) \cosh(\lambda_j + \eta)} \frac{Q(\lambda_j - 2\eta) Q(\lambda_j + 4\eta + \mathrm{i}\pi)}{Q(\lambda_j + 6\eta) Q(\lambda_j + \mathrm{i}\pi)}, \quad j = 1, \dots, M. \tag{21}
$$

172  Although the $U(1)$ symmetry remains broken, the system exhibits an additional sym-
173  metry under Eq. (17), which ensures the existence of a homogeneous $T$-$Q$ relation. In this
174  case, the integer $M$ is fixed by the system parameters. One can then find a proper "local
175  vacuum" to proceed with the generalized Bethe ansatz approach and study the physical
176  quantities of the model [26,28]. When $M \ge 2N$, the $T$-$Q$ relation (18) with $M$ Bethe roots
177  may provide the complete set of eigenvalues of the transfer matrix. When $0 \le M < 2N$,
178  two $T$-$Q$ relations are required to parameterize the full spectrum of the transfer matrix,
179  with the number of Bethe roots being $M$ and $2N - M - 1$, respectively. Such degen-
180  erate points exist in various integrable models, e.g., the anisotropic spin-$\frac{1}{2}$ chains with
181  non-diagonal boundary fields [26, 28–30, 35]).

182  **Diagonal boundaries**   As the boundary parameter $\epsilon$ approaches infinity $\epsilon \to +\infty$, the
183  resulting $K$-matrices become diagonal

$$
K^-(u) = \mathbb{I}, \quad K^+(u) = \mathcal{M}, \tag{22}
$$

184  where the matrix $\mathcal{M}$ is defined in (3). In this case, the $K$-matrices automatically satisfy
185  the operator relation

$$
\lim_{\eta \to 0} \{K^+(u) K^-(u)\} = \mathbb{I}. \tag{23}
$$

186  The $U(1)$-symmetry of the IK model is now recovered, and one can also use homoge-
187  neous $T$-$Q$ relations to parameterize the spectrum of the transfer matrix. The functions
188  $\mathbf{a}(u)$, $\mathbf{b}(u)$ and $\mathbf{d}(u)$ given by Eqs. (8)-(10) reduce to [47]

$$
\bar{\mathbf{a}}(u) = \prod_{l=1}^{N} c(u - \theta_l) c(u + \theta_l) \frac{\sinh(u - 6\eta) \cosh(u - \eta)}{\sinh(u - 2\eta) \cosh(u - 3\eta)}, \tag{24}
$$

$$
\bar{\mathbf{d}}(u) = \prod_{l=1}^{N} d(u - \theta_l) d(u + \theta_l) \frac{\sinh u \cosh(u - 5\eta)}{\sinh(u - 4\eta) \cosh(u - 3\eta)}, \tag{25}
$$

$$
\bar{\mathbf{b}}(u) = \prod_{l=1}^{N} b(u - \theta_l) b(u + \theta_l) \frac{\sinh u \sinh(u - 6\eta)}{\sinh(u - 2\eta) \sinh(u - 4\eta)}. \tag{26}
$$

189  The $T$-$Q$ relation (12) now can be simplified as the following one [47]

$$
\Lambda(u) = \bar{\mathbf{a}}(u) \frac{Q(u + 4\eta)}{Q(u)} + \bar{\mathbf{d}}(u) \frac{Q(u - 6\eta + \mathrm{i}\pi)}{Q(u - 2\eta + \mathrm{i}\pi)} + \bar{\mathbf{b}}(u) \frac{Q(u - 4\eta) Q(u + 2\eta + \mathrm{i}\pi)}{Q(u - 2\eta + \mathrm{i}\pi) Q(u)}, \tag{27}
$$

where the function $Q(u)$ is defined as

$$Q(u) = \prod_{j=1}^{M} \sinh\left(\frac{u - \lambda_j - 2\eta}{2}\right) \sinh\left(\frac{u + \lambda_j - 2\eta}{2}\right).$$

(28)

In this case, the integer $M$ is adjustable and can take the following values

$$M = 0, 1, \ldots, N.$$

(29)

The resulting BAEs for the Bethe roots $\{\lambda_j\}$ in (27) are

$$\prod_{l=1}^{N} \frac{\sinh\left(\frac{\lambda_j - \theta_l - 2\eta}{2}\right) \sinh\left(\frac{\lambda_j + \theta_l - 2\eta}{2}\right)}{\sinh\left(\frac{\lambda_j - \theta_l + 2\eta}{2}\right) \sinh\left(\frac{\lambda_j + \theta_l + 2\eta}{2}\right)} \frac{\sinh(\lambda_j - 2\eta)\cosh(\lambda_j + \eta)}{\sinh(\lambda_j + 2\eta)\cosh(\lambda_j - \eta)}$$

$$= -\frac{Q(\lambda_j - 2\eta)Q(\lambda_j + 4\eta + i\pi)}{Q(\lambda_j + 6\eta)Q(\lambda_j + i\pi)}, \qquad j = 1, \ldots, M.$$

(30)

In the following sections, we will construct integrable IK Gaudin models with open boundaries and derive their exact spectra.

# 5  IK Gaudin model with open boundaries

Following the approach outlined in Section 3.1, one can construct the associated Gaudin operators $\{H_j\}$. We expand the transfer matrix $t(\theta_j)$ around $\eta = 0$, specially as follows

$$t(\theta_j) = \kappa\, t_0(\theta_j)(\mathbb{I} + \eta H_j + \cdots), \quad j = 1, \ldots, N,$$

$$H_j = \left.\frac{\partial \ln(t(\theta_j)/\kappa)}{\partial \eta}\right|_{\eta=0}.$$

(1)

Equation (7) implies that

$$t_0(\theta_j) = \lim_{\eta \to 0} \mathrm{tr}_0 \left\{ \prod_{l \neq j}^{N} \sinh(\theta_j - \theta_l) \prod_{l=1}^{N} \sinh(\theta_j + \theta_l) K_0^+(\theta_j) P_{0,j} K_0^-(\theta_j) \right\}$$

$$= \prod_{l \neq j}^{N} \sinh(\theta_j - \theta_l) \prod_{l=1}^{N} \sinh(\theta_j + \theta_l) \lim_{\eta \to 0} \{ K_j^-(\theta_j) K_j^+(\theta_j) \}.$$

(2)

To ensure that the resulting Gaudin operators form a commuting family, i.e.,

$$[H_i, H_j] = 0, \qquad i, j = 1, 2, \ldots, N,$$

which is essential for the integrability of the corresponding Gaudin model [1], we require that $t_0(\theta_j)$ be proportional to the identity operator (see Section 3.1)

$$\lim_{\eta \to 0} \left\{ K_j^-(\theta_j) K_j^+(\theta_j) \right\} \propto \mathbb{I} \quad .$$

(3)

Equation (3) gives rise to the following restrictions for the boundary parameters

$$\lim_{\eta \to 0} e^{\sigma} = e^{\sigma'}, \quad \lim_{\eta \to 0} e^{\epsilon'} = -e^{\epsilon}.$$

(4)

203 Without loss of generality, we assume that the boundary parameters $\epsilon$, $\epsilon'$ and $\sigma$ are
204 independent of the crossing parameter $\eta$, while $\sigma'$ depends on $\eta$. From these assumptions,
205 we get the following identities

$$\sigma' = \sigma + \bar{\sigma}\eta, \quad \epsilon' = \epsilon + i\pi. \tag{5}$$

206 As a consequence, the following equation can be obtained

$$\mathsf{t}_0(\theta_j) = \prod_{l \neq j}^{N} \sinh(\theta_j - \theta_l) \prod_{l=1}^{N} \sinh(\theta_j + \theta_l)\, w(\theta_j) \times \mathbb{I}, \quad w(\theta_j) = (1 - 4e^{-2\epsilon}\sinh^2\theta_j). \tag{6}$$

207 Using the initial condition of $R$-matrix (4) and the QYBE (3), the double row transfer
208 matrix at the point $u = \theta_j$ can be expressed as [32]

$$
\begin{aligned}
t(\theta_j) = {}& \kappa R_{j,j-1}(\theta_j - \theta_{j-1}) \cdots R_{j,1}(\theta_j - \theta_1) K_j^-(\theta_j) R_{1,j}(\theta_j + \theta_1) \cdots R_{j-1,j}(\theta_j + \theta_{j-1}) \\
& \times R_{j+1,j}(\theta_j + \theta_{j+1}) \cdots R_{N,j}(\theta_j + \theta_N) \mathrm{tr}_0\{K_0^+(\theta_j)P_{0,j}R_{j,0}(2\theta_j)\} \\
& \times R_{j,N}(\theta_j - \theta_N) \cdots R_{j,j+1}(\theta_j - \theta_{j+1}).
\end{aligned} \tag{7}
$$

209 Then, the following Gaudin operator can be constructed

$$H_j = \Gamma_j(\theta_j) + \sum_{l \neq j}^{N} \bar{\Gamma}_{j,l}(\theta_j, \theta_l), \tag{8}$$

210 where the operator $\Gamma_j(\theta_j)$ and $\bar{\Gamma}_{j,l}(\theta_j, \theta_l)$ are defined as

$$\Gamma_j(\theta_j) = \frac{1}{\sinh(2\theta_j)w(\theta_j)} \lim_{\eta \to 0} \frac{\partial}{\partial \eta}\Big[ K_j^-(\theta_j)\, \mathrm{tr}_0\{K_0^+(\theta_j)P_{0,j}R_{j,0}(2\theta_j)\} \Big], \tag{9}$$

$$\bar{\Gamma}_{j,l}(\theta_j, \theta_l) = \frac{r_{j,l}(\theta_j - \theta_l)}{\sinh(\theta_j - \theta_l)} + \frac{K_j^-(\theta_j)r_{l,j}(\theta_j + \theta_l)K_j^+(\theta_j)}{w(\theta_j)\sinh(\theta_j + \theta_l)}\bigg|_{\eta \to 0}, \tag{10}$$

211 and the operator $r_{j,l}(u)$ is given by (4) and (9). Here $\Gamma_j(\theta_j)$ describes the on-site potential,
212 while $\bar{\Gamma}_{j,l}(\theta_j, \theta_l)$ represents a site-dependent, long-range two-site interaction. One can also
213 use the spin-1 operators to expand the Gaudin operator. After some tedious calculations,
214 we get the expression of $\Gamma_j(u)$

$$
\begin{aligned}
\Gamma_j(u) = {}& -\frac{e^{-\sigma}\sinh u\,(4e^u - e^{\epsilon}(\bar{\sigma}+2))}{e^{2\epsilon} - 4\sinh^2 u}(S_j^-)^2 - \frac{e^{\sigma}\sinh u\,(4e^{-u} + e^{\epsilon}(\bar{\sigma}+2))}{e^{2\epsilon} - 4\sinh^2 u}(S_j^+)^2 \\
& -\frac{4\sinh(2u)}{e^{2\epsilon} - 4\sinh^2 u}(S_j^z)^2 - \frac{4\sinh u\,(e^{\epsilon} - (\bar{\sigma}+2)\sinh u)}{e^{2\epsilon} - 4\sinh^2 u}S_j^z \\
& -\frac{e^{2\epsilon}(5 + 7\cosh(2u)) + 5 + 4\cosh(2u) - 9\cosh(4u)}{\sinh(2u)\left(e^{2\epsilon} - 4\sinh^2 u\right)}\mathbb{I}_j.
\end{aligned} \tag{11}
$$

215 The two-site interaction $\bar{\Gamma}_{j,l}(u)$ comprises not only the operator $r_{i,j}(u)$ given in Eq. (9),
216 but also two additional operators on site $j$

$$
\begin{aligned}
K_j^-(u)|_{\eta \to 0} = {}& e^{-\sigma - \epsilon}\sinh u\,(S_j^-)^2 + e^{\sigma - \epsilon}\sinh u\,(S_j^+)^2 \\
& + 2e^{-\epsilon}\sinh u\,(S_j^z)^2 + \left(1 - 2e^{-\epsilon}\sinh u\right)\mathbb{I}_j, \tag{12} \\
K_j^+(u)|_{\eta \to 0} = {}& -e^{-\sigma - \epsilon}\sinh u\,(S_j^-)^2 - e^{\sigma - \epsilon}\sinh u\,(S_j^+)^2 \\
& - 2e^{-\epsilon}\sinh u\,(S_j^z)^2 + \left(1 + 2e^{-\epsilon}\sinh u\right)\mathbb{I}_j. \tag{13}
\end{aligned}
$$

Unlike the two-site interaction $\Gamma_{j,l}(\theta_j, \theta_l)$ in the periodic system (see Eq. (8)), $\bar{\Gamma}_{j,l}(\theta_j, \theta_l)$ in the open system depends on the inhomogeneous parameters $\theta_{j,l}$ and the boundary parameters $\sigma$ and $\epsilon$.

The Gaudin operator defined in Eq. (8) is exactly solvable. Equation (1) allows us to obtain the eigenvalues of the Gaudin operators from the exact spectrum of the transfer matrix $t(u)$ at $u = \theta_j$ as follows

$$E_j = \left. \frac{\partial \ln(\Lambda(\theta_j)/\kappa)}{\partial \eta} \right|_{\eta=0}. \tag{14}$$

As demonstrated in Section 4, the homogeneous $T$-$Q$ relation exhibits a significantly simpler form compared to its inhomogeneous counterpart, yielding BAEs that are more tractable for analytical and numerical treatment. Consequently, in the following section, we will consider the exact solutions of the IK Gaudin model with boundary conditions specified by Eqs. (17) and (22), respectively.

# 6 Exact solution of the IK Gaudin model with open boundaries

## 6.1 Constrained non-diagonal boundaries

Following Eqs. (17) and (4), one can construct the corresponding Gaudin operator (8) by letting $\bar{\sigma} = -4\mathbf{k}$ and $\mathrm{e}^{\epsilon'} = -\mathrm{e}^{\epsilon}$. It should be noted that the extra constraint (17) only affects the specific expression of the operator $H_j$, without altering its underlying structure (the position of non-zero entries in the matrix).

The eigenvalues of the IK Gaudin operators read

$$\begin{aligned}
E_j &= \left. \frac{\partial \ln(\Lambda(\theta_j)/\kappa)}{\partial \eta} \right|_{\eta=0} \\
&= \sum_{l=1}^{M} \frac{4 \sinh \theta_j}{\cosh \theta_j - \tilde{\mu}_l} - 6 \coth \theta_j - \tanh \theta_j + \frac{4\mathrm{e}^{-2\epsilon} \sinh(2\theta_j)}{1 - 4\mathrm{e}^{-2\epsilon} \sinh^2(2\theta_j)} \\
&\quad + \sum_{k \neq j}^{N} \left[ \frac{1 - 5\cosh(\theta_j - \theta_k)}{\sinh(\theta_j - \theta_k)} + \frac{1 - 5\cosh(\theta_j + \theta_k)}{\sinh(\theta_j + \theta_k)} \right],
\end{aligned} \tag{1}$$

and the corresponding BAEs are

$$-\sum_{k \neq j}^{M} \frac{\tilde{\mu}_j + 3\tilde{\mu}_k}{\tilde{\mu}_j^2 - \tilde{\mu}_k^2} + \sum_{l=1}^{N} \frac{1}{\tilde{\mu}_j - \cosh \theta_l} + \frac{4\tilde{\mu}_j}{\mathrm{e}^{2\epsilon} + 4 - 4\tilde{\mu}_j^2} = 0, \qquad j = 1, \ldots, M. \tag{2}$$

Notably, there is a one-to-one correspondence between $\tilde{\mu}_j$ in Eq. (2) and $\lambda_j$ in Eq. (21): $\tilde{\mu}_j = \lim_{\eta \to 0} \cosh \lambda_j$. From Eqs. (1) and (2), we observe that the eigenvalue of the Gaudin operator $H_j$ depend on the set $\{\tilde{\mu}_j\}$ and $\epsilon$, but is independent of $\sigma$ and $\bar{\sigma}$. The numerical solutions of Eq. (2) for small-scale systems are presented in Table 2.

## 6.2 Diagonal open boundaries

When $\epsilon \to +\infty$, the expression of the Gaudin operator (8) can be simplified as follows

$$\Gamma_j(\theta_j) = (-6\coth \theta_j - \tanh \theta_j) \times \mathbb{I}, \tag{3}$$

$$\Gamma'_{j,l}(\theta_j, \theta_l) = \frac{r_{j,l}(\theta_j - \theta_l)}{\sinh(\theta_j - \theta_l)} + \frac{r_{l,j}(\theta_j + \theta_l)}{\sinh(\theta_j + \theta_l)}. \tag{4}$$

The corresponding eigenvalues of the IK Gaudin operators in terms of the Bethe roots

| $\tilde{\mu}_1$ | $\tilde{\mu}_2$ | $\tilde{\mu}_3$ | $E_1$ | $d$ |
|---|---|---|---|---|
| 1.0462 | −31.3108 | 0.0541 | −17.5222 | 1 |
| 0.1573 | 1.0461 | – | −17.5011 | 2 |
| 1.3552 | 1.0504 | – | −16.3699 | 2 |
| −1.5098 | 1.0453 | – | −15.3492 | 2 |
| 1.0478 | 1.3151 | −1.2458 | −11.8797 | 1 |
| −1.1210 | 1.0471 | 1.1846 | −1.9845 | 1 |
| 0.2520−0.7199i | 0.2520+0.7199i | – | 28.9411 | 2 |
| −8.7360 | −1.6847 | – | 30.4393 | 2 |
| 0.8132−0.3340i | 0.8132+0.3340i | 1.3385 | 32.7812 | 1 |
| −0.2126 | 1.3237 | 14.4331 | 36.8243 | 1 |
| −0.4131 | 1.3235 | – | 36.8791 | 2 |
| 2.0772 | 1.3188 | – | 39.7611 | 2 |
| 27.7924 | −0.0643 | 1.1818 | 46.1381 | 1 |
| −0.1616 | 1.1820 | – | 46.1579 | 2 |
| −1.4080 | 1.1716 | – | 48.6891 | 2 |
| 1.5999 | 1.1923 | – | 49.1452 | 2 |
| 1.2134−0.0445i | 1.2134+0.0445i | −1.3308 | 52.8331 | 1 |

Table 2: Numeric solutions of Eq. (2) with $N = 3$, $\{\theta_1, \theta_2, \theta_3\} = \{-0.40, 0.18, 0.67\}$ and $\{\epsilon, \sigma, \bar{\sigma}\} = \{0.50, 0.12, -4\}$. From Eq. (20), the number of Bethe root $M$ can be 2 or 3. The eigenvalue of $H_1$ given by Eq. (1) matches the exact diagonalization results. Here $d$ represents the degeneracy of the energy level.

are

$$E_j = \sum_{k=1}^{M} \frac{4 \sinh \theta_j}{\cosh \theta_j - \bar{\mu}_k} - \tanh \theta_j - 6 \coth \theta_j$$
$$+ \sum_{k \neq j}^{N} \left[ \frac{1 - 5 \cosh(\theta_j - \theta_k)}{\sinh(\theta_j - \theta_k)} + \frac{1 - 5 \cosh(\theta_j + \theta_k)}{\sinh(\theta_j + \theta_k)} \right], \tag{5}$$

where $\{\bar{\mu}_j | j = 1, \ldots, M\}$ satisfy the following BAEs

$$\sum_{l=1}^{N} \frac{1}{\bar{\mu}_j - \cosh \theta_l} - \sum_{k \neq j}^{M} \frac{\bar{\mu}_j + 3\bar{\mu}_k}{\bar{\mu}_j^2 - \bar{\mu}_k^2} = 0, \qquad j = 1, \ldots, M. \tag{6}$$

Analogously, $\bar{\mu}_j$ in Eq. (6) and $\lambda_j$ in Eq. (30) has a one-to-one correspondence $\bar{\mu}_j = \lim_{\eta \to 0} \cosh \lambda_j$. The numerical solutions of Eq. (6) for small-scale systems are presented in Table 3.

# 7 Conclusions

In this paper, we study the integrable IK Gaudin model with both periodic and open boundary conditions. We construct the Gaudin operator by expanding the inhomogeneous

| $\bar{\mu}_1$ | $\bar{\mu}_2$ | $\bar{\mu}_3$ | $E_1$ | $d$ |
|:---:|:---:|:---:|:---:|:---:|
| 1.5573 | – | – | $-44.7405$ | 5 |
| $-0.3537$ | 1.5480 | – | $-43.6642$ | 3 |
| – | – | – | $-41.2908$ | 7 |
| $0.2509-0.6787\mathrm{i}$ | $0.2509+0.6787\mathrm{i}$ | – | $-38.9182$ | 3 |
| 1.5348 | 1.0476 | $-1.2797$ | 4.8930 | 1 |
| 1.0480 | – | – | 8.3428 | 5 |
| 0.3834 | 1.0468 | – | 8.9485 | 3 |

Table 3: Numeric solutions of Eq. (6) with $N = 3$, $\{\theta_1, \theta_2, \theta_3\} = \{0.40, 0.18, 1.20\}$. The eigenvalue of $H_1$ given by Eq. (6) matches the exact diagonalization results. Here $d$ represents the degeneracy of the energy level.

transfer matrix at the point $u = \theta_j$, which ensures the solvability of the Gaudin operator. By leveraging the exact solutions of the IK model, obtained through the Bethe-ansatz method, we finally get the eigenvalue spectrum of the Gaudin operator in terms of the Bethe roots, which are determined by the corresponding Bethe ansatz equations (BAEs). Numerical computations have also been done to validate our analytical results.

The IK model with open boundary conditions deserves further elaboration. In this paper, we only consider the non-diagonal $K$-matrices in Eqs. (1) and (2). It should be emphasized that our method remains applicable to other $K$-matrix classes examined in Refs. [45, 46]. Since the Gaudin operator's form explicitly depends on boundary parameters, distinct $K$-matrix configurations will result in physically distinct Gaudin operators.

Future work may involve further analysis of BAEs. It will be interesting to explore the existence of infinite Bethe roots or singular physical solutions of the corresponding BAEs, which may be essential for the completeness of our BAEs and the explanation of the degeneracy.

Another open question is the construction of eigenstates for the Gaudin operator. When the system retains $U(1)$ symmetry, the eigenstates can be constructed using the algebraic Bethe ansatz method. However, this approach requires further adaptation or generalization to address the case withou $U(1)$ symmetry.

# Acknowledgments

Financial support from the National Natural Science Foundation of China (Grant Nos. 12105221, 12205335, 12247103, 12074410, 12047502), the Strategic Priority Research Program of the Chinese Academy of Sciences (Grant No. XDB33000000), Shaanxi Fundamental Science Research Project for Mathematics and Physics (Grant Nos. 22JSZ005), the Scientific Research Program Funded by Shaanxi Provincial Education Department (Grant No. 21JK0946), Beijing National Laboratory for Condensed Matter Physics (Grant No. 202162100001), and the Double First-Class University Construction Project of Northwest University is gratefully acknowledged.

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
