# Peer review of "Exact solution of the Izergin-Korepin Gaudin model with periodic and open boundaries"

_SciPost Physics Core_

## Round 1 · Referee Report · Jules Lamers (Referee 1) · 2025-1-7

Strengths
1- Gaudin models associated to the Izergin--Korepin (IK) R-matrix of the twisted affine algebra A_2^{(2)} with periodic and with open boundary conditions are introduced.
2- A Bethe-ansatz description of their spectrum is given.
Weaknesses
1- The new results in this paper are incremental.
2- The significance of the new models is not motivated enough.
3- Neither physical nor mathematical properties are discussed.
4- The literature is not cited appropriately.
Report
The authors introduce Gaudin models associated to the Izergin-Korepin R-matrix of the twisted affine algebra A_2^{(2)} with periodic and with open boundary conditions. The Bethe-ansatz description of their spectrum is given, based on previous work using the 'off-diagonal Bethe ansatz' by several of the authors and their collaborators. In the case of open boundaries, the model simplifies somewhat if the K-matrix satisfies a condition ensuring that the TQ equation has no inhomogeneous term and in the special case of diagonal reflection, where U(1) (i.e. spin z) is conserved, the boundary term simplifies significantly.
For SciPost Physics, unfortunately I do not see how this paper meets any of the publication criteria. Compared to previous works, the new results in this paper are incremental, and based on straightforward calculations. Indeed, technically, the results just amount to the computation of a semiclassical limit of said R-matrix, of the resulting inhomogeneous periodic and double-row transfer matrices, and of the (known) eigenvalues and Bethe ansatz equations. Neither physical properties (structure of solutions to Bethe ansatz equations, resulting spectrum, correlations, ...) nor mathematical properties (completeness? dualities?) are discussed, nor is the importance of the model motivated sufficiently.
For SciPost Physics Core, I would at the very least ask for revisions, cf the following.
Requested changes
1- The significance of the results should be argued much more convincingly.
1a- Is the main interest physical, mathematical, or methodological? Any of these should be clearly motivated.
1b- If no Gaudin model without U(1) symmetry has been studied before, why not start with untwisted type A? If they have been studied, what is the expected added value of this particular case; where might it to show up in physics?
2- The literature is not cited appropriately.
2a- The introduction and references are heavily biased towards previous works of the authors and their collaborators, which are cited rather extensively compared to the rest of the literature.
2b- The many important recent developments for Gaudin models (completeness, dualities, etc) are completely missing.
2c- It is not clearly stated whether any Gaudin models without U(1) symmetry have been studied before, and relevant literature should be cited.
3- The terms 'integrability' and 'exact solvability' are used too sloppily. Given the highly developed state of affairs for other Gaudin models, clear and precise statements should be given about the integrability / exact solvability of the new models.
3a- The term 'integrable' is mentioned in the second paragraph of Section 1, but not explained.
3b- In Section 3, it is shown that the N evaluations of the transfer matrix at the different inhomogeneity parameters have trivial classical limit, and thus their semiclassical limits H_j still commute. However, one moreover ought to show that these are independent. For various other Gaudin models, nowadays there are very precise statements available whether the 'Gaudin subalgebra' is maximal commuting.
3c- Later in Section 3, it is concluded from the commutativity of the H_j that the model is exactly solvable. I agree this would be expected, but even if the Gaudin algebra could be shown to be maximal (which would require work) I am not sure this would by itself prove that the model is exactly solvable.
3d- Next the authors invoke their prior results of the off-diagonal Bethe ansatz to give a description of the eigenvalues in terms of Bethe-ansatz equations. To conclude the exact solvability from this would still require a proof of completeness. For other Gaudin models, this has been shown.
3e- In Remark 6.1, it is mentioned that 'M Bethe roots may provide the complete set of eigenvalues of the transfer matrix.' Is this an expectation? Is there numerical evidence that this is the case? Can it be proven?
4- There are various typos that should be corrected.
4a- One typo is in the affilation b.
4b- Another recurrent typo is that in math mode, like $T-Q$, '-' becomes a minus sign, which is incorrect.
4c- In several places, a space is missing (e.g. twice on page 2).
4d- Just below (3.8), 'inhomogeneous' should be 'inhomogeneity'.
4e- Just below (5.10), subscripts are mis-formatted (twice).
4f- In ref [2] there is a typo in the title.
5- Some other things should be clarified.
5a- Section 1, second paragraph, "The lack of ... novel Gaudin models." is unclear. What is meant by 'many' integrable models? What is meant by 'always'? How is the first part of the sentence related to 'studying novel Gaudin models'?
5b- Two paragraphs below, the phrase 'non-A-type' seems to suggest the IK model is not of type A, which is not true. The meaning of the sentence should be clarified.
5c- End of Section 1, two times: why are 'exact solutions' plural? Is this a typo, or intended; in the latter case this should be explained.
5d- In (2.6), it is not explained what V_1 is.
5e- Just below, the term 'quantum transfer matrix' is potentially confusing, cf the work of Göhmann, Klümper and collaborators.
5f- In (2.8), explain that the superscript '(p)' is for periodic.
5g- Below (2.9), as mentioned above, the term 'proved' is not quite correct.
5h- For the Bethe ansatz equations (2.15) a citation should be given.
5i- As usual, (3.3) follows directly from (2.4)-(2.8), unlike the current phrasing suggests. This could be clarified.
5j- In (3.8), what is the reason to keep the prefactor (3.7) in the expression? It could have been factored out in (3.1).
5k- The proof on the bottom of page 6 and top of page 7 is obvious from expanding (2.9) in \eta and using (3.7). This could be clarified before the proof.
5l- The paragraph following the proof breaks the flow of the text. Consider turning it into a Remark or moving it to Section 5.
5m- Does the proof of (4.7) also rely on the crossing relation? Any other properties?
5n- Can (5.9) and (5.10) be simplified? It is rather unsatisfactory as a result.
5o- Where did the condition (6.5) appear previously? Provide the key citations if possible.
Recommendation
Accept in alternative Journal (see Report)

---

## Editorial Decision

accepted_in_target_journal